# Measuring Floating Thick Seep Oil from the Coal Oil Point Marine Hydrocarbon Seep Field by Quantitative Thermal Oil Slick Remote Sensing

**Ira Leifer** [1,*], **Christopher Melton** [1], **William J. Daniel** [1], **David M. Tratt** [2], **Patrick D. Johnson** [2], **Kerry N. Buckland** [2], **Jae Deok Kim** [1] **and Charlotte Marston** [1]

1   Bubbleology Research International, Inc., Solvang, CA 93463, USA; christopher.melton@bubbleology.com (C.M.); bill.daniel@bubbleology.com (W.J.D.); jaedeokkim314@gmail.com (J.D.K.); charlotte.marston@bubbleology.com (C.M.)
2   The Aerospace Corporation, El Segundo, CA 90245, USA; david.m.tratt@aero.org (D.M.T.); patrick.d.johnson@aero.org (P.D.J.); kerry.n.buckland@aero.org (K.N.B.)
*   Correspondence: ira.leifer@bubbleology.com

**Abstract:** Remote sensing techniques offer significant potential for generating accurate thick oil slick maps critical for marine oil spill response. However, field validation and methodology assessment challenges remain. Here, we report on an approach to leveraging oil emissions from the Coal Oil Point (COP) natural marine hydrocarbon seepage offshore of southern California, where prolific oil seepage produces thick oil slicks stretching many kilometers. Specifically, we demonstrate and validate a remote sensing approach as part of the Seep Assessment Study (SAS). Thick oil is sufficient for effective mitigation strategies and is set at 0.15 mm. The brightness temperature of thick oil, $T_{BO}$, is warmer than oil-free seawater, $T_{BW}$, allowing segregation of oil from seawater. High spatial-resolution airborne thermal and visible slick imagery were acquired as part of the SAS; including along-slick "streamer" surveys and cross-slick calibration surveys. Several cross-slick survey-imaged short oil slick segments that were collected by a customized harbor oil skimmer; termed "collects". The brightness temperature contrast, $\Delta T_B$ ($T_{BO} - T_{BW}$), for oil pixels (based on a semi-supervised classification of oil pixels) and oil thickness, $h$, from collected oil for each collect provided the empirical calibration of $\Delta T_B(h)$. The $T_B$ probability distributions provided $T_{BO}$ and $T_{BW}$, whereas a spatial model of $T_{BW}$ provided $\Delta T_B$ for the streamer analysis. Complicating $T_{BW}$ was the fact that streamers were located at current shears where two water masses intersect, leading to a $T_B$ discontinuity at the slick. This current shear arose from a persistent eddy down current of the COP that provides critical steering of oil slicks from the Coal Oil Point. The total floating thick oil in a streamer observed on 23 May and a streamer observed on 25 May 2016 was estimated at 311 (2.3 bbl) and 2671 kg (20 bbl) with mean linear floating oil 0.14 and 2.4 kg m$^{-1}$ with uncertainties by Monte Carlo simulations of 25% and 7%, respectively. Based on typical currents, the average of these two streamers corresponds to 265 g s$^{-1}$ (~200 bbl day$^{-1}$) in a range of 60–340 bbl day$^{-1}$, with significant short-term temporal variability that suggests slug flow for the seep oil emissions. Given that there are typically four or five streamers, these data are consistent with field emissions that are higher than the literature estimates.

**Keywords:** Coal Oil Point seep field; oil remote sensing; thermal infrared; petroleum emissions; emulsion

## 1. Introduction

### 1.1. Floating Marine Oil

There are two primary sources of floating marine oil: natural marine seepage and anthropogenic releases, including consumption from terrestrial run-off and atmospheric deposition, transportation-related (accidental) releases, and pipeline and tanker discharges [1]. Oil transport safety has increased over recent decades—leading to significant decreases in spilled oil from significant spills [2]. Still, accidents continue to happen and merit the

best technology available to mitigate damage to the coastal and marine ecosystem [3,4], economy [5,6], and human health [7]. The most critical spill response need is for thick oil maps—with thick oil defined as amenable to mitigation by in situ burning or oil booming, skimming, and collection. Thick slicks are defined as 0.5–3 mm depending on oil type, weathering, approach, etc. [8,9]. Here, remote sensing provides the requisite fast turnaround given the data's time sensitivity [10] with quantitative [11] and semi-quantitative [12,13] thickness remote sensing approaches demonstrated.

### 1.2. Study Motivation

The lack of quantitative field validation remains a key factor impeding the further use of remote sensing in oil spill response. The Seep Assessment Study (SAS) demonstrates an in-scene calibration of oil thickness thermal contrast remote sensing for thick floating oil. Here, we operationally define thick oil as actionable. This definition differs from the traditional definition, which relates to visible appearance. In the SAS, short oil slick streamer segments are sensed remotely and physically collected for later quantification. Each collected segment is termed a "collect". Analysis of the collects provided an empirical model that related oil thickness to oil temperature brightness contrast, $\Delta T_B$, with oil-free sea surface. This function was applied to the along-slick streamer imagery to derive floating oil mass. SAS leveraged the perennial thick oil slicks from the Coal Oil Point (COP) natural marine hydrocarbon seep field, located in waters offshore California, a critical advantage given the infeasibility of planned releases in US waters.

$\Delta T_B$ is the difference between the observed oil brightness temperature, $T_{BO}$, and the oil-free water brightness temperature, $T_{BW}$, which was provided by a discontinuous spatial model. The $T_{BW}$ model was discontinuous to account for distinct water masses on the slick's opposite sides. The discontinuity arises from a persistent eddy downcurrent (westwards) of Coal Oil Point, an oceanographic feature proposed by Leifer [14] to play an important role in transporting COP seep field oil slicks. Airborne hyperspectral thermal infrared (TIR) imagery collected by the Mako sensor clearly showed the eddy, including fine-scale structure. These data also showed the eddy trapping oil slicks.

Extension to thin oil must address oil slicks with $T_{BO}$ close to $T_{BW}$, which is beyond this study's scope and arises from the lower emissivity of oil than water which impacts $T_B$. Specifically, thin oil analysis needs to reference the true temperature contrast, $\Delta T$, and thus correctly applying emissivity, which describes non-blackbody thermal behavior. Therefore, a non-TIR approach is required to classify oil pixels from water pixels.

### 1.3. Marine Hydrocarbon Seepage

The SAS leveraged natural seep oil emissions from the COP seep field. Natural seepage is the dominant marine oil source, with global emissions estimated at 200,000–2,000,000 tons $yr^{-1}$ and 80,000–240,000 tons $yr^{-1}$ in North America [1]. The wide range arises because there are few published natural seep emission estimates [14] partly due to the challenge of oil emissions quantification at sea and a lack of approaches to quantify such emissions. Here, too, remote sensing can play an important role. For example, MacDonald, et al. [15] used remote sensing to estimate Gulf of Mexico emissions of $2.5$–$9.4 \times 10^4$ m$^3$ yr$^{-1}$, equivalent to 22,500–85,500 tons yr$^{-1}$.

Natural marine hydrocarbon seepage is the migration of geological oil and gas from a reservoir to the seabed and into the ocean through faults and fractures [16]. Marine hydrocarbon oil seepage provides a natural oil spill science laboratory [14], including remote sensing science [17]. Marine oil seepage occurs in all major ocean basins and continental shelves [18].

Most seep emissions manifest as widely dispersed oil and gas emissions that create very thin surface sheens [15]. One notable exception is the shallow (2–70 m water depths) Coal Oil Point (COP) seep field [14], where thick oil slicks are a perennial feature [19]. COP seepage escapes from several square kilometers of the seabed [20] as oily and non-oily bubbles and bubble plumes [21]. Oil emissions were estimated at 100 bbl day$^{-1}$

in 1995 based on a sonar survey and scaling by the oil to gas ratio for the largest seep in the field [22]. The Hornafius, Quigley and Luyendyk [22] estimate was similar to the 1970 estimate of 50–70 bbl oil day$^{-1}$ based on aerial imagery and assumed oil slick thickness by Allen, et al. [23]. Notably, though, gas emissions were at a cyclical minimum in 1995 [24] and peaked in 2010, declining since [25], suggesting the potential for significant cyclical trends in oil emissions.

Seep gas emissions are highly variable on a range of time and length scales [14] from the sub-hourly [26] to decadal Leifer [14], including episodic emissions. Episodic oil emissions can be slug flow of an oil cap followed by increasing gas and decreasing oil emissions until both eventually subside [27]. Episodic emissions can impose a "head" on the oil slick that eventually diminishes until it disappears (Leifer, unpublished personal observation, Nov. 2016).

### 1.4. Marine Oil Slick Evolution

COP seep field oil slicks form near the seabed source and evolve under weathering, dispersion, and transport processes [19] on timescales from hours to weeks [28]. Oil slick advection is from winds and currents, with the oil drifting at the vector sum of a fraction of the current and wind speed (referenced to 10-m). These fractions are termed the windage factor, $W_F$, typically 2%–3% [29] and a current factor, $C_F$. In a COP study, Leifer, et al. [30] found that oil drift could be explained by $W_F$ = 12% and $C_F$ = 0 or a $W_F$ = 3% and $C_F$ > 0.

Although several slick processes—spreading, turbulence, etc., suggest oil slicks should disperse [28], slicks generally tend to aggregate due to Langmuir cells [29], current convergence zones, and current shears. Current shears can arise from bathymetric effects [31] and other processes. Typical wind-driven oil slicks are asymmetric [29] due to the competition between gravitational spreading and surface tension spreading, which opposes the tendency of oil to "bunch up" [31].

Marine oil slicks thicker than sheen primarily exist as stable emulsions—mixtures of microscopic water and oil droplets. Turbulence energy in the oil slick forms stable emulsions, typically 70%–80% water [32]. Emulsions form when oil asphaltene compounds encounter the interface of tiny water droplets in the oil and form solid and stable films around the droplets that stabilize the emulsion [32]. Emulsions dramatically increase the oil's viscosity and decrease its evaporation, affecting physical and optical properties and processes such as weathering [33].

### 1.5. Oil Slick Thermal Infrared and Visible Remote Sensing

Underlying $\Delta T$ is the significantly different optical depths of oil compared to seawater. As a result, the oil absorbs solar insolation near the air-oil interface versus the upper several meters for clear oil-free seawater [34]. According to Beer's Law, absorbed radiation decreases exponentially with distance with a length-scale termed the extinction length, $\tau$,

$$F(z) = F_0 e^{-z/\tau}, \tag{1}$$

where $F(z)$ is the radiation flux at depth, $z$, and $F_0$ is the insolation at the oil surface (Figure 1).

This absorbed energy flows within the oil slick and to the overlying air and underlying water, adding to the overall heat flow between the warmer air and cooler water (typical daytime conditions). Thus, $\Delta T$ results from the balance between these heat flows, the sea surface radiation balance, and heat flow in the overlying air and underlying water. These energy flows, in turn, depend on the oil's optical and physical parameters, including thermal conductivity, specific heat capacity, and emissivity, which are different from seawater properties.

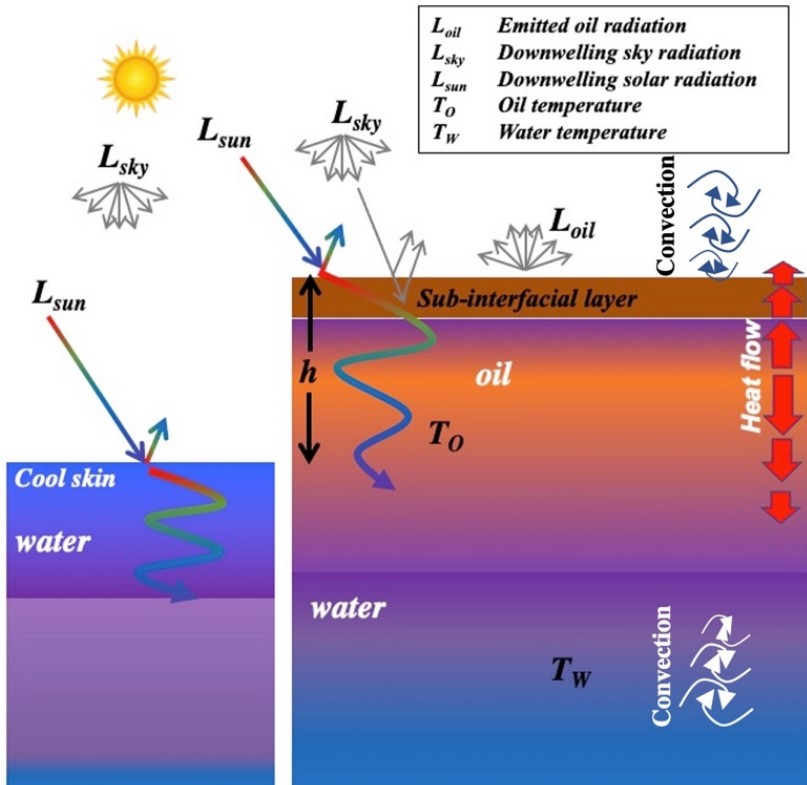

**Figure 1.** Oil slick thermal energy balance schematic illustrating the relationship between thickness, *h*, and oil-water thermal contrast.

The differential surficial heating between water and oil (and differing emissivities) allows TIR remote sensing to detect oil by $\Delta T_B$ [35–39], with the physical and thermal properties affecting $\Delta T_B$ [39–41].

The near-surface air was warmer than the water for the SAS study, leading to downwards heat flux under conditions of intense solar insolation. For optically thin oil slicks, most of the energy transits the oil into the underlying seawater. For optically thick oil slicks, the incident radiation largely is absorbed. As a result, $\Delta T$ increases little with further increases of slick thickness, *h*, causing $\Delta T_B(h)$ to asymptote in the thick oil limit.

For a TIR sensor, the relationship between surface temperature, *T*, and $T_B$ depends on emissivity $\varepsilon$. For a blackbody, $T_B = T$, i.e., $\varepsilon$, for other surfaces, $\varepsilon < 1$ [42] and is defined by the Stefan-Boltzmann Law [43],

$$E_B = \varepsilon\sigma T^4,\tag{2}$$

where $E_B$ is the blackbody emissive power, $\sigma$ is the Stefan-Boltzmann constant, and *T* is in kelvin. For a TIR sensor in the real world, the reflected downwelling thermal radiance, $L_{sky}$, must be subtracted from the at-sensor radiance, $L_S$,

$$L_S = \zeta\sigma\varepsilon T_B^4 = \varepsilon\sigma\zeta T_S^4 + (1-\varepsilon)\zeta L_{sky},\tag{3}$$

where $\zeta$ is atmospheric transmittance. Crude oil emissivity, $\varepsilon_o$, for thick oil slicks varies between 0.93 and 0.97 depending on wavelength and oil type [44]. Niclòs, et al. [45] found an angular dependency in $\varepsilon_o$, decreasing from 0.956 to 0.875 for viewing angles from 15° to 65°. For reference, seawater emissivity, $\varepsilon_w$, varies from 0.9831 to 0.9767 for 8.0 to 13 µm for overhead viewing and 0.9821 to 0.9881 for 30° off-angle viewing for 8.0 to 13 µm, respectively [46].

A key implication of oil emissivity being less than that of water is that $\Delta T_B < 0$ if $\Delta T = 0$; an infinitely thin oil slick is the limiting case. As *h* decreases towards an infinitely thin slick, $T_O$ decreases towards $T_W$ while oil emissivity increases towards that of seawater,

"warming" $T_B$. Thus, there is an oil thickness transition, $h_T$, where $\Delta T_B = 0$. Several studies suggest $h_T$ is 50–150 μm [39,47] although Grierson [48] observed warm oil sheens as thin as 1-μm for a night survey. This very thin $h_T$ (<1 μm) for nighttime observations suggest a complex relationship between $h_T$ and environmental conditions. For example, Lu, Zhan and Hu [39] showed that $h_T$ depends on insolation and thus varies with time of day.

Solving Equation (2) for $T$ yields:

$$T = \sqrt[4]{\frac{T_B^4 - (1 - \varepsilon)T_{sky}^4}{\varepsilon}}, \tag{4}$$

where $T_{sky}$ is the sky temperature. Typical $T_{sky}$ for a clear mid-latitude sky is 245 K and 225 K for summer and winter, respectively [49]. These temperatures are far colder than typical sea surface temperatures and thus contribute significantly less reflected radiation than surface thermal emissions (e.g., Equation (2)). Low clouds are warmer, inducing significant corrections.

For this study, the focus on thick oil allows basing the analysis on $T_B$. Furthermore, the in-scene calibration allows neglection of $L_{sky}$, which affects "collects" and streamer observations equally.

## 2. Methods

### 2.1. Overview

The study collected sea surface and airborne data using SeaSpires™, an oil spill mapping science package with the methodology described in Leifer, et al. [50]. SeaSpires acquired airborne VIS and TIR imagery, orientation, and position information of along-slick streamer surveys and cross-slick "collect" surveys. Collects were short segments of the oil slick streamer that were boomed, skimmed, and offloaded into storage containers for later oil volumetric quantification. Three boats facilitated the collects, two of which were boom vessels and an offload vessel. Skimming used a customized harbor oil skimmer. Collects provided an in-scene calibration of the relationship between oil thickness to TIR thermal contrast between the collected oil slick and oil-free seawater. The calibration function is applied to along-slick remote sensing data to map spatial patterns in the floating oil mass.

Airborne imagery was pixel geo-registered based on airplane position and orientation data and was analyzed to derive the brightness temperature contrast, $\Delta T_B$, between oil-free water, $T_{BW}$, and oil, $T_{BO}$. Two approaches were used to calculate $\Delta T_B$, a probability distribution approach for the (single scene) collects and a model of $T_{BW}$ across the slick.

The collects provided an oil thickness, $h$, calibration function relating $h$ to $\Delta T_B$. Specifically, the average oil pixel's $\Delta T_B$ in the boom was related to $h$, which was derived from the collected mass, $M$, the oil density, $\rho$, and the oil pixels' area. The empirical model is:

$$\Delta T_B(h) = \chi\left(1 - e^{h/\tau}\right), \tag{5}$$

where $\chi$ is the thick oil limit of $\Delta T_B$, for which there is no additional energy to absorb from the incoming radiation. In the thin limit ($h = 0$), $\Delta T_B = 0$. The function includes an exponential by analogy with Beer's law. The empirical model is based on a least-squares linear regression analysis.

### 2.2. Oil Collection

The approach to derive the calibration function comprised two components—oil collection for volume/thickness assessment and remote sensing analysis. In brief, two 33-m long, 12.5-cm (6") diameter harbor booms are connected to both sides of a modified weir skimmer by two ~10 m tow ropes to the surface vessels, *F/V Maalea* and *F/V Rock Steady*. The collection protocol reduced entrainment to negligible levels by treating the oil gently during booming. Specifically, the boom vessels towed the boom very slowly and smoothly (gently) and asymmetrically (to offset the skimmer from the boom apex) across

an oil streamer segment, guided by the airplane. After boom loading, the boom is closed and adjusted to symmetric for offloading. Closing the booms rolls the oil at the boom apex, which incorporates air into the oil, stabilizing the captured slick against entrainment. A gasoline-powered water pump offloaded oil into 20-L plastic buckets on the *F/V Double Bogey*. Where needed, hand offloading used a pool leaf skimmer. Buckets were weighed back at the laboratory (NTEP Ranger 7000, Ohaus Corp., Parsippany, NJ, USA, 35 kg, 0.5 g accuracy; LC3200D, Sartorius, Göttingen, Germany, 3 kg, 1 mg accuracy), referenced with Class 1 calibration weights (Troemner, Thorofare, NJ, USA).

### 2.3. Remote Sensing Analysis

#### 2.3.1. Overview

A nadir-looking airplane-mounted system collected visible and thermal (specifically, microbolometer) imagery of the sea surface, including oil slicks. We also analyzed hyperspectral thermal imagery acquired in 2013 by the Mako sensor on a Twin Otter airplane.

The $T_B$ enhancement is calculated in two manners. For collects, $T_B$ probability distributions are computed for the scene and modeled to derive the reference water $T_B$. A semi-supervised approach classifies thermal imagery pixels as thin or thick oil, seawater, wake, boom, and boat. For streamer imagery, sea surface water $T_B$ outside the oil slick is modeled to account for gradients. The model allows for discontinuities at the slick. Then, the $T_B$ contrast is relative to the modeled (oil-free) sea surface, i.e., if the oil was absent.

#### 2.3.2. Microbolometer and Visible Remote Sensing Acquisition

The SeaSpires' core is two visible, VIS, spectrum video cameras, and one TIR video camera. The visible cameras are a high-resolution 30-megapixel video camera (7K HDPro, Avigilon, Allen, TX, USA), denoted HDPro, and a wide-angle 1-megapixel video camera (1.0MP-HD-DN, Avigilon, TX, USA), which provided flight-targeting guidance. A research-grade TIR camera (A655sc, FLIR, Nashua, NH, USA), denoted A655, recorded $640 \times 480$ pixel resolution TIR video. Although closely aligned, video image analysis revealed a slight angular offset between the 7K and A655sc cameras, corrected during geo-registration. A portable computer records video on a RAID array displays video for monitoring video data and helping guide flight lines to target specific oil slick features. The portable computer provides control signals for the video cameras and records other data, such as position and orientation.

For this study, SeaSpires was configured for installation in the wheel well of a Cessna 2015 airplane. An inertial navigation system (INS) (VN-300, VectorNav, Dallas, TX, USA) mounted to the airplane cockpit dashboard provided geo-location and orientation for airborne surveys. A Global Positioning System (GPS) receiver (19X, Garmin) provided geo-location for boat deployments. Video cameras were nadir-looking through ports in a customized cowl to prevent wind vibrations. A guide camera provided wide field-of-view imagery to align flight lines for collects.

SeaSpires improvements for marine vessel deployment include a GPS time server (1000A, Time Machines Corp., Lincoln, NE, USA) and upwards and downwards UV-NIR spectrometers (HR4000 Ocean Optics, Orlando, FL, USA), and downwelling and upwelling visible and TIR radiation sensors matched to the TIR camera and spectrometers (Apogee Instruments, Logan, UT, USA). SeaSpires data includes meteorology measurements (3D sonic anemometer, humidity, temperature) for boat deployments.

#### 2.3.3. Imagery Geo-Registration

The first step is imagery geo-registration to account for platform orientation and motion and minor alignment mismatch between the TIR and VIS cameras. TIR and VIS images were geo-registered to overlay oil features accurately, a process described in Leifer, et al. [51]. In brief, before geo-registration, data are de-spiked (median filter to identify spike points for interpolation), and the dropped data points are interpolated in position and orientation. Additionally, time-base irregularities were corrected. Finally,

a five-point (0.1 s) median filter with nearest-neighbor-averaging filtered noise, yielding ~0.02 m uncertainty. Video pixels are re-gridded to 0.2-m resolution using 2D cubic-spline interpolation, significantly greater than uncertainty. Geo-registration used the Image GeoRectification And Feature Tracking (ImGRAFT) toolbox (MATLAB2016, Mathworks, Nantick, MA, USA).

### 2.3.4. Brightness Temperature Contrast: Collects

$\Delta T_B$ is relative to the collection scene for water brightness temperature ($T_{BW}$), which is derived from the scene's $T_B$ probability distribution (Figure 2B). Specifically, scenes contain several elements, including the significantly warmer boat and oil boom, oil (thick and thin), cooler, undisturbed water, and much colder boat-wake water. Thresholding allows easy segregation of the boat and oil boom pixels.

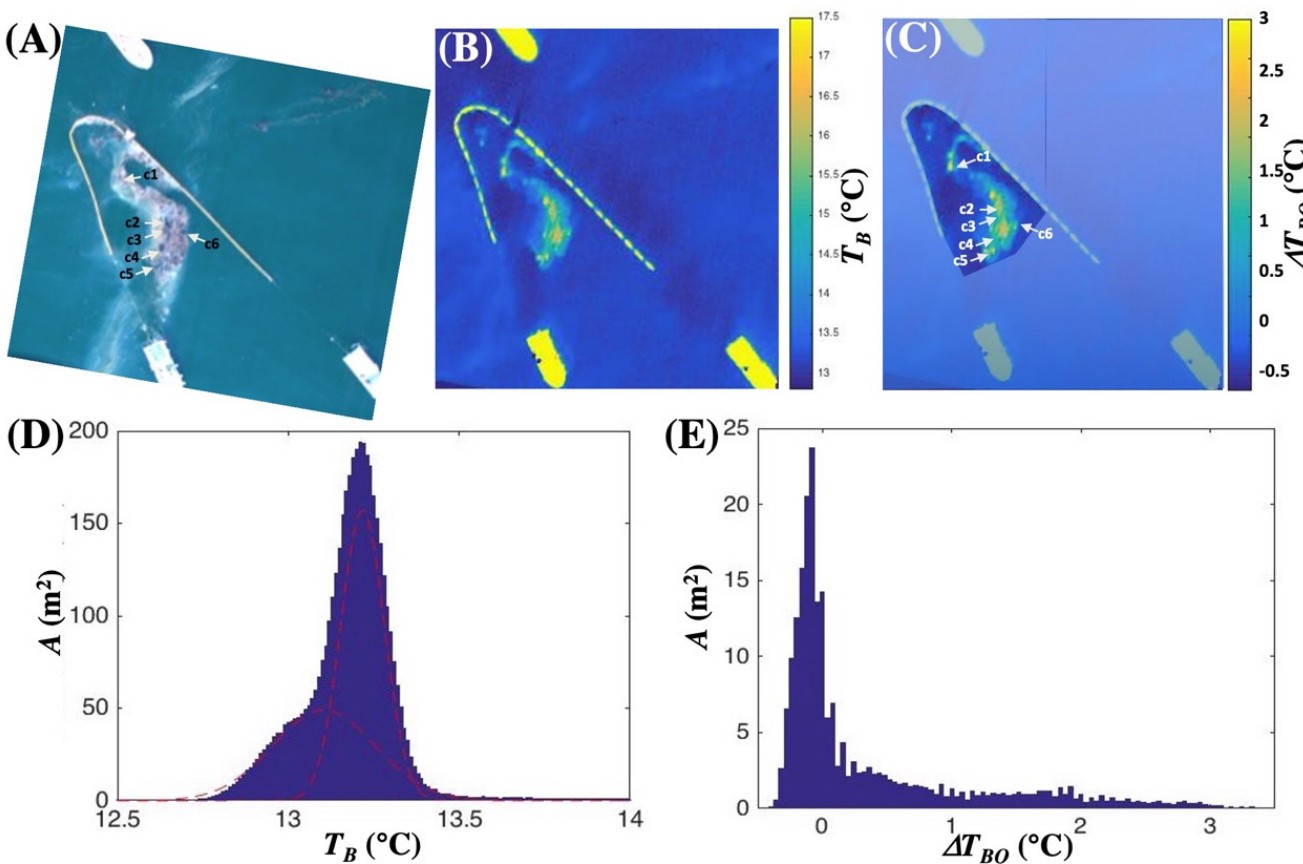

**Figure 2.** Oil collection #10 for (**A**) Visible, black oil identified. (**B**) Brightness temperature, $T_B$, image. (**C**) Oil brightness temperature contrast ($\Delta T_{BO}$) image, the area outside boom dimmed. Visible black oil locations labeled. (**D**) Area ($A$) histogram of $T_B$ (number × pixel area) and Gaussian curve fit to model water scene elements. (**E**) Area histogram of $\Delta T_{BO}$ for the boom image subset shown in (**C**). Data key on the figure. Imagery acquired 23 May 2016.

Two Gaussian functions modeled the water probability distribution for the non-wake, $\Phi_W$, and wake, $\Phi_{W1}$, water-scene elements (Figure 2D,E). $\Delta T_B$ is calculated relative to $\Phi_W$.

Specifically, a broad distribution ($\Phi_O$) modeled the oil temperature probability distribution, and two narrower distributions modeled the oil-free water, $\Phi_W$, and the wake, $\Phi_{W1}$,

$$\Phi = \Phi_O + \Phi_w + \Phi_{w1} = a_O\, e^{-\frac{(T_{BP\_o} - T_B)^2}{W_o^2}} + a_w e^{-\frac{(T_{BP\_w} - T_B)^2}{W_w^2}} + a_{w1} e^{-\frac{(T_{BP\_w1} - T_B)^2}{W_{w1}^2}}, \quad (6)$$

where $a_O$, $a_W$, and $a_{W1}$ are the amplitudes or maxima of the distributions, $\Phi_O$, $\Phi_W$, and $\Phi_{W1}$, for the oil and water masses, respectively, with distribution half-widths, $W_O$, $W_W$,

and $W_{W1}$ and peaks at $T_{BP\_O}$, $T_{BP\_W}$, and $T_{BP\_W1}$. $T_B$ probability distribution functions, $\Phi$, in each image were modeled by three Gaussian functions using the curve fitting toolbox (cftool.m, MATLAB2016, Mathworks, Nantick, MA, USA).

First, pixels inside the boom are segregated spatially. Then, the area coverage distribution $A(T_{BO})$, is derived from the histogram of $T_{BO}$ and is the product of the pixel area and the number of pixels in each $\Delta T_{BO}$ bin.

$A(T_B)$ shows a strong, narrow Gaussian shape with a peak at $T_{BW}$ from the boom water pixels with $\Delta T_{BO}$ up to 3K. These small patches of thick, warm oil largely match visible oil patches that appeared black (subset Figure 2C), but not completely—some of the black oil pixels do not exhibit a strong $T_B$ anomaly (e.g., Figure 2C(c3,c4,c5)). Meanwhile, the hotspot between c4 and c6 shows no visible distinguishing features.

### 2.3.5. Brightness Temperature Contrast: Surveys

Water brightness temperature ($T_{BW}$) for along slick surveys was more challenging to derive as oil slicks stretched for several kilometers, covering oceanographic structures that impose sea surface temperature gradients, including at the intersection of two distinct water masses. Where the two water masses meet, there is a temperature discontinuity.

The sea surface $T_{BW}$ was modeled to account for the discontinuity and gradients at the slick, described in Supplementary Section S3. In brief, $T_{BW}(Y)$ is determined from a combination of a linear polynomial and a sinusoidal function fit to non-outlier water pixels, i.e.,

$$T_{BW}(Y) = b + cY + \sin(gY), \tag{7}$$

where $b$, $c$, and $g$ are fit parameters, and $Y$ is transverse or cross-slick distance. $T_{BW}(Y)$ is a two-part piecewise linear function that was discontinuous across the slick. Thus, a Gaussian transition function filled the gap across the slick described by Supplementary Section S3.2. The brightness temperature contrast profile, $\Delta T_B(Y)$, relative to $T_{BW}(Y)$, is

$$\Delta T_B(X, Y) = T_B(X, Y) - T_{BW}(X, Y), \tag{8}$$

where $X$ is along-slick distance and uses a linear trend with $X$ between image subsets (which have ~90% overlap) with overlapping $\Delta T_B(X, Y)$ values between adjacent image subsets averaged.

### 2.3.6. Empirical Thickness Model

The empirical model (Equation (5)) relates the collected oil mass, $M$, to $A(\Delta T_B)$ and is applied to the $\Delta T_{BO}$ of the oil slicks streamers. The model implementation is a two-step iteration,

$$M(\Delta T_{BO}) = k(\Delta T_{BO}) \int_0^\infty 1 * A(\Delta T_{BO}) d\Delta T_{BO} + N$$

$$M(\Delta T_{BO}) = k_2(\Delta T_{BO}) \int_0^\infty k(\Delta T_{BO}) A(\Delta T_{BO}) d\Delta T_{BO} + N, \tag{9}$$

where $k$ is the initial scaling parameter (set to unity) and $N$ is the error. Then, $k_2$ is calculated based on $M(\Delta T_{BO})$ from all the collects by minimizing $N$.

$M$ was determined by weighing buckets with oil, subtracting the empty bucket and lid weights, $792 \pm 8.5$ g and $197.1 \pm 1.3$ g, respectively, and the amount of unemulsified water. Then, bucket sub-samples were centrifuged to separate the oil and water to provide the emulsification level. The oil fraction in the collected oil to water emulsion ratio was ~90%. The empirical model forces the best fit to pass through $M = 0$ where $\Delta T_{Bo} = 0$ within the 70-mK measurement uncertainty.

Monte Carlo simulations (10,000 simulations) provided the basis for estimating uncertainty with variability or error in $T_B$ used in the $\Delta T_B$ calculation based on the standard deviation of the sea surface temperature (0.084 K). This was taken as the uncertainty in each of the collect measurements that underlay the calibration function. Notably, this is similar to sensor uncertainty. Each simulation calculated a new calibration function, which

then was applied to the data, with the probability distribution function of the resultant floating oil mass fit by a Gaussian function with a half-width that defined the uncertainty.

### 2.3.7. Hyperspectral Thermal Infrared Acquisition and Analysis

The airborne Mako instrument acquired hyperspectral thermal infrared imagery of the COP seep field on 29 August 2013 from an altitude of 3.7 km, yielding a ground resolution of ~2 m. Mako is a broad-area whiskbroom scanning spectral imager that has participated in numerous atmospheric composition and solid Earth studies. Hall, et al. [51] and Buckland, et al. [52] comprehensively describe its technical details and capabilities.

Spectral analysis of the COP imagery used the procedures detailed in Buckland et al. [52]. In brief, the spectroradiometrically-calibrated and atmospherically-compensated data were processed utilizing the standard adaptive coherence estimation (ACE) approach. This procedure found that the water aerosol spectrum in the spectral library used (there is no spectrum for the oil available) consistently identified oil slick streamers on the ocean surface. Furthermore, the oil slick streamers on the ocean surface were characterized by a conspicuous spectral feature centered on 9.5 μm. Figure 3A is a false-color TIR radiance image of the scene where RGB coding renders the oil in a red hue. Five markers along a transect intersecting three prominent streamers denote locations where example spectra were extracted (Figure 3B). To visualize variances better, each spectral channel in Figure 3B has been de-medianed, i.e., median-subtracted. This clearly exposes the 9.5-μm feature associated with the oil (ID2, ID3, ID4) compared to the sea surface without thick oil (ID1, ID5) and also illustrates the oil's influence on the broader spectrum, which shares an affinity with the water aerosol spectrum. This observation explains why the ACE algorithm returns a water aerosol identification for the oil.

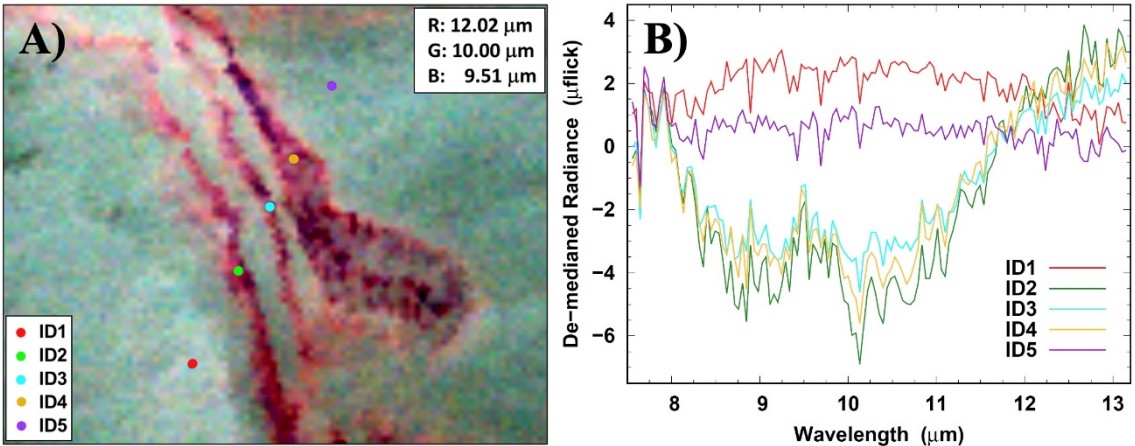

**Figure 3.** Spectral behavior of COP oil slick. (**A**) RGB coded radiance image with five spectral sampling locations shown. (**B**) De-medianed spectra extracted from sampling locations in panel (**A**).

Brightness temperature, $T_B$, maps are derived from the atmospherically-compensated radiance data, $L_C$, by first using the standard Inverse Planck Function treatment to calculate a wavelength-dependent $T_B$ (K) for each pixel $(i, j)$:

$$T_B(\lambda, \ L_C) = cd_2 \left[ \lambda \ ln \left( \frac{d_1}{\lambda^5 \ L_C(i,j,\lambda)} + 1 \right) \right]^{-1}, \tag{10}$$

where coefficients $d_1$ and $d_2$ are $1.19104 \times 10^{10}$ μW sr$^{-1}$ cm$^{-2}$ μm$^{-4}$ and $1.43877 \times 10^4$ K μm, respectively, $L_C$ is in units of μflick (μW sr$^{-1}$ cm$^{-2}$ μm$^{-1}$), and $\lambda$ is wavelength (μm). Following this, the median value is computed over a specified wavelength interval, usually in the range $8.0 < \lambda < 12.5$ μm, to exclude spectral regions where the atmospheric compensation is less reliable and thus provide for a more robust estimate of $T_S(i, j)$ by substituting the median of $T_B(\lambda, L_C)$.

This technique for estimating brightness temperature has proven reliable during previous tests over many years, where ground-truth data on temperatures are available for comparison. The thermometric resolution (Noise Equivalent Differential Temperature, NEDT) of these data is ~30 mK.

## 3. Results

### 3.1. Setting

The study focused on oil slicks from the COP seep field in the northern Santa Barbara Channel, California. The COP seep field is located in water from a few meters to ~85 m deep (oily seepage arises from deeper than 20 m), extending to ~3 km offshore and covering 6.3 km² [14,25]. The COP seep field comprises many focused seep areas—regions of high spatial vent density, separated by larger areas with sparse to no seepage [53]. The spatial distribution follows geological structures along the offshore Elwood Trend, the inshore COP trend, and an Inshore Trend [14]. The latter seep trend includes the IV Super Seep and is not associated with slicks (Figure 4A).

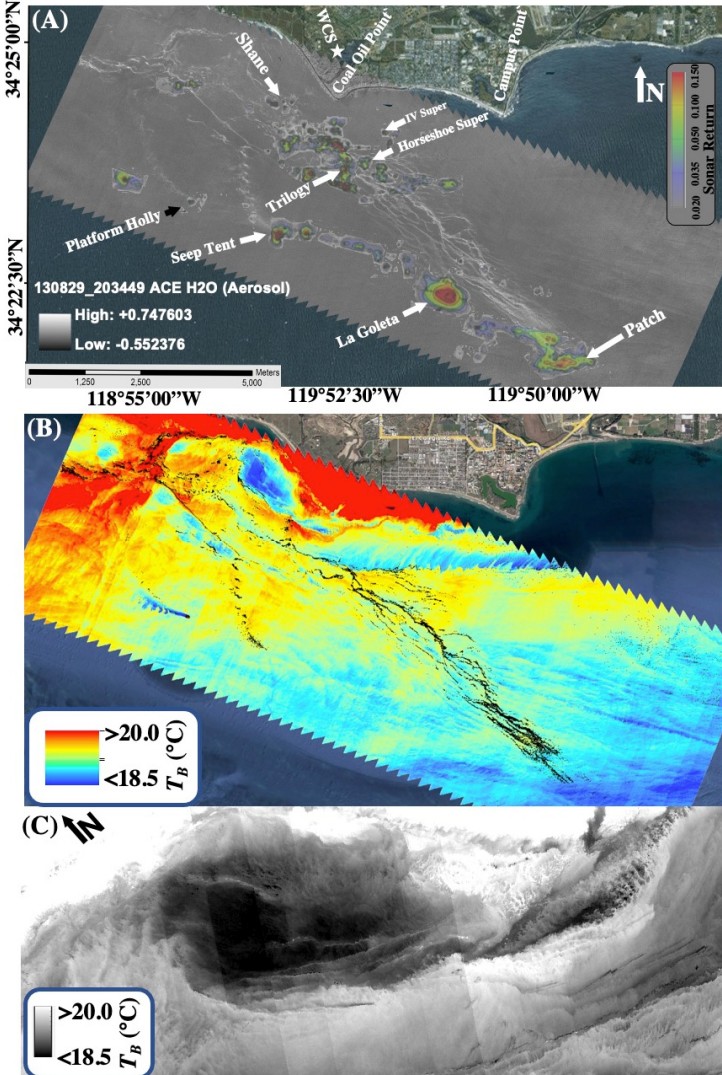

**Figure 4.** (**A**) ACE filter imagery of the COP seep field and sonar seepage map—adapted from Leifer [14]. Major, informally named seep areas labeled. WCS is West Campus Station, from which wind data were acquired. (**B**) Brightness temperature map and ACE oil detection (black). (**C**) COP gyre detail—see Supplementary Figure S1 for full $T_B$ map. Data key on panel (**A**). Imagery acquired 29 August 2013. Panels (**A**) and (**B**) shown in the Google Earth environment.

Each separate seep area produces distinct (and not necessarily continuous) oil slicks or "streamers", forming parallel streamers under typical daytime wind and current conditions (Figure 4A). These manifest as discrete waves of beach tar stranding as the streamers are driven ashore by the sea breeze [54]. The most prolific seep oil source in the Mako data (29 August 2013) was the Patch Seep, located at the southeastern edge of the field (Figure 4). Slicks from Patch Seep converge with slicks from the La Goleta Seep area and drift into the COP trend seeps, which also feature prolific oil emissions. This drift direction and speed result from winds and currents, particularly surface currents, which are strongly related to winds [19]. The linear slick morphology reflects convergence motions and significant gradients in winds across the field with relatively strong east-southeasterly winds to the east and southerly and weak in the west, where slicks arc to the north. Patches of cooler and warmer waters are evident throughout the scene, representing areas of divergence (cooler), convergence (warmer), and current shears—thermal structures related to flow fields that play a role in oil slick trajectories.

Overall currents align with the coastline, which shifts orientation significantly at COP, and underlies significant features in $T_B$ (Figure 4B). Perhaps most apparent is the gyre to the west of COP (a persistent feature) [14,55], which separates cooler inshore waters from warmer offshore waters. This gyre or eddy is a Lagrangian coherent structure (LCS) that is cooler in the center (Figure 4C), suggesting upwelling with divergence at the gyre edges (likely driving a three-dimensional circulation against the coastline and elsewhere). A weak current shear also originates from Campus Point, flowing westwards primarily at the edge of the kelp beds off Isla Vista. A second current shear also follows the Isla Vista coastline, ~750 m offshore, and exhibits ripples with size scales of a few hundred meters. Back projection suggests this current shear could have originated from the Santa Barbara coastal headlands. This current shear veers with the coastline to the west of COP, following the gyre towards the shore.

In the absence of strong winds, the eddy likely transports slicks towards the coast, a few kilometers west of COP. Typical afternoon sea breeze winds impose an onshore drift component that directs the slicks closer to Coal Oil Point [14] with kelp beds providing a natural study endpoint. Thus, the study area focused on a few kilometers WNW of the COP seep field to provide sufficient experimental drift time.

### 3.2. Environmental Conditions

On 23 May 2016, winds were weak and from the south, i.e., typical for coastal marine stratified conditions in the early to late morning, with the planetary boundary layer (PBL) beginning to grow at 0800. Winds were from West Campus Station (Figure 4) and began rising towards afternoon strength around 1400 (Supplementary Figure S2). On 25 May 2016, the PBL started growing earlier (~07:00 LT) with winds strengthening continuously over the morning, reaching white-capping strength circa noon. Around noon on the field survey days, wave heights were 0.83 and 0.58 m; wave periods were 5.26 and 6.25 s on 23 and 25 May 2016, respectively. Current radar data (CODAR) suggested that morning currents were offshore out of the north, which shifted towards the east midday and then flowed to the northwest in the late afternoon (Supplementary Figure S3).

### 3.3. In-Scene Calibration

The calibration function (Equation (5); Figure 5) was developed based on two constraints: in the thick-oil limit, no additional energy is available for absorption, thus corresponding to an asymptote, whereas in the zero-oil limit ($h = 0$), $\Delta T_B = 0$. Allowing the zero-oil limit to vary within the TIR measurement uncertainty, 0.07K, improved the least-squares linear-regression analysis, i.e., improved $R^2$.

**Table 1.** Details on collects.

| Collect | Day, Time | $M$ | Mean $T_B$ |
|---|---|---|---|
| (#) | (-) | (kg) | (°C) |
| 9 | 23 May 08:54 | 51.67 | 1.05 |
| 10 | 23 May 10:37 | 51.84 | 1.87 |
| 16 | 25 May 09:58 | 10.57 | 0.12 |
| 17 | 25 May 10:30 | 7.92 | 0.08 |

$T_B$—brightness temperature, $M$—collected oil mass.

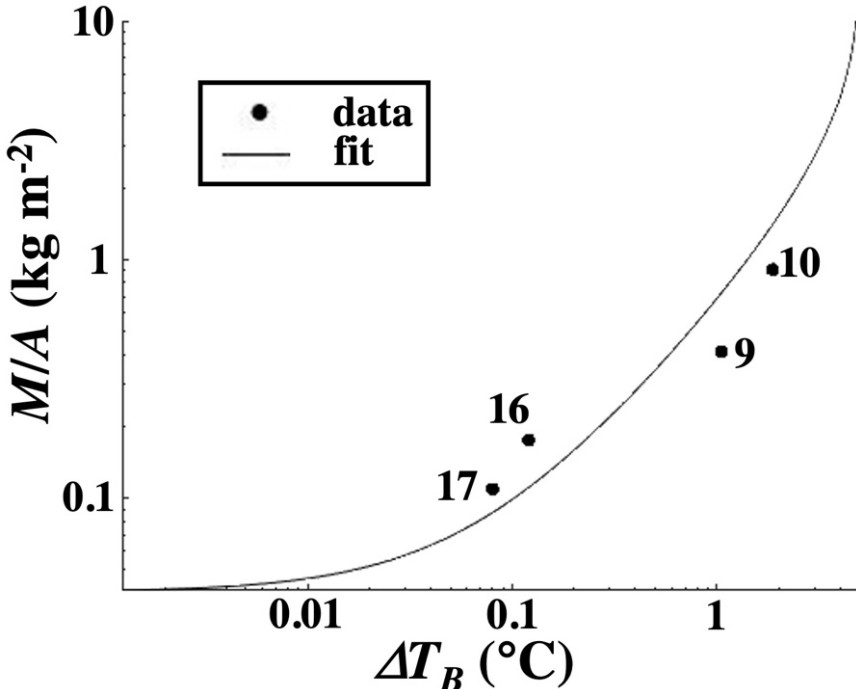

**Figure 5.** Brightness temperature contrast, $\Delta T_B$, versus the mass, $M$, to area, $A$, ratio, for calibration "collects" and asymptotic exponential fit. Data key and collect number (details in Table 1) on figure.

*3.4. Sea Surface and Slick Thermal Structure*

Airborne TIR and VIS imagery were acquired at altitudes primarily from 150 to 250 m with flight lines aligned either perpendicular to the oil streamer during oil collection (booming) or along an oil streamer while surface vessels were offloading and setting up for the following collect. Orthogonal flight lines provided aerial data for the in-scene calibration. TIR spatial resolution was ~15 cm; visible imagery resolution was ~5 cm.

The oil slick meanders as it drifts downcurrent (Figure 6) along the intersection where offshore, warmer water meets cooler nearshore water (see $\Phi$ for two different water masses in Figure 4C and Supplementary Figure S1). The along-slick survey data show warm pools are common, e.g., at $X$ = 100, 500, 1100, 1650, and 2100 m (Figure 6C). Each of these warmer pools corresponds to areas of oil aggregation as indicated by strong negative $\Delta T_B$.

This warm pool creates a "discontinuity" or very sharp north-south gradient across the slick (Figure 7B). Notably, this pool does not correspond to visible sheens, which are apparent to the west of the pool (Figure 6A). The modeled water brightness temperature, $T_{BW}(X, Y)$, showed the warmer water pool with a relatively sharp, shoreward edge stretching alongside the slick (Figure 7C).

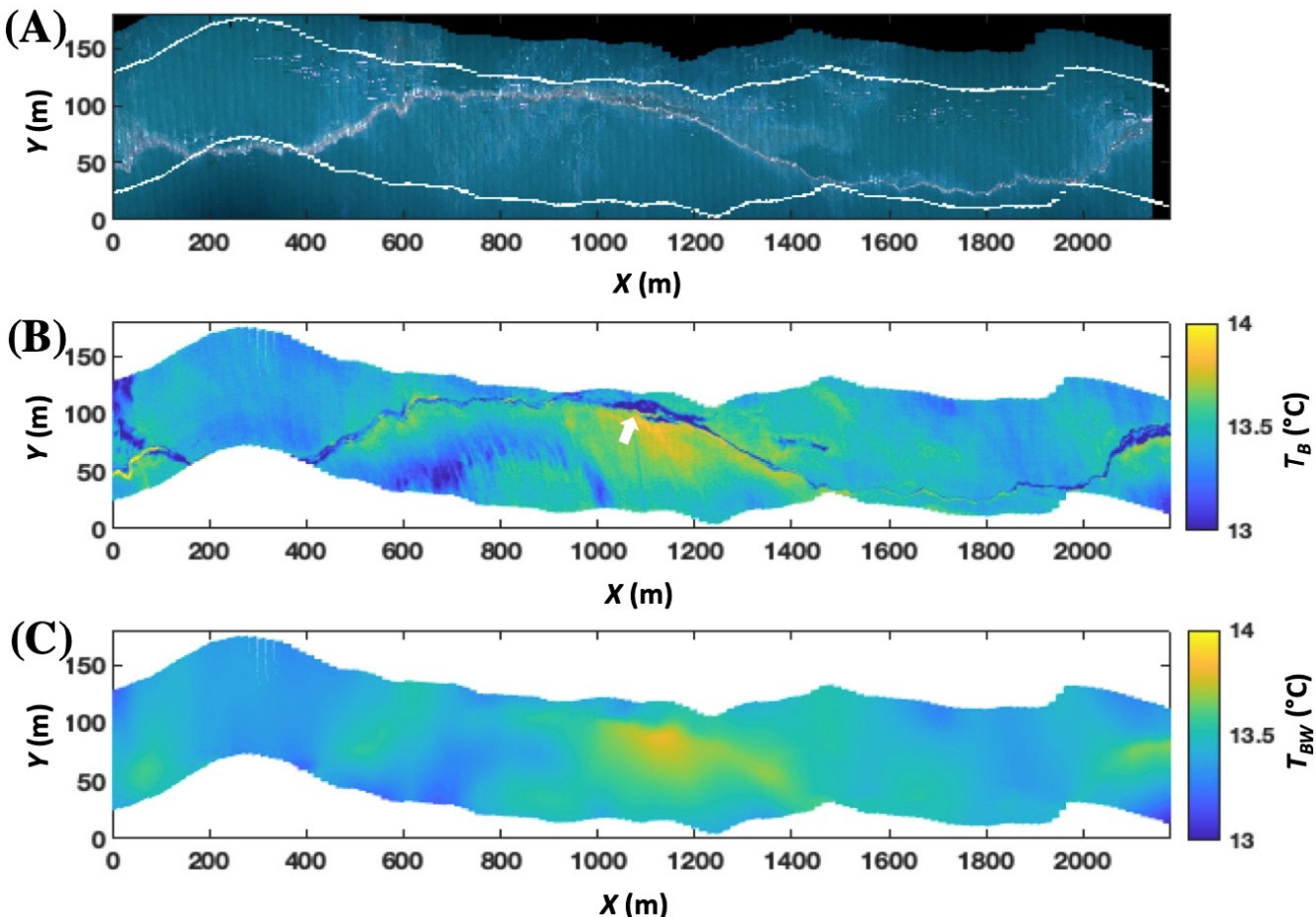

**Figure 6.** (**A**) Visible image photomosaic of an oil slick for 23 May in the along-slick and cross-slick coordinates, (*X*) and (*Y*), respectively. (**B**) Mosaicked thermal infrared image of sea surface brightness temperature ($T_B$) projected to a 20-cm grid. The white arrow identifies the location of the image mosaic in Figure 7A. (**C**) Water brightness temperature, $T_{BW}(X, Y)$, respectively. Data key on figure.

The slick location at the front between the two water masses seems an unlikely coincidence, more likely relating to surface convergence. The warm pool is also elongated in the drift axis direction. Interestingly, the offshore warmer water shows a warming trend towards the slick, suggesting warmer water "piling up" or convergence. Note that $T_{BW}$ indicates this warm pool is unrelated to the vessel's wake, which would mix cooler water to the surface. The absence of the boat in the visible image suggests the wake is old.

The thermal and visible imagery reveal significant fine-scale structure within the oil slick. For example, the slick shows four separate thick oil streams in the visible (Figure 8), which shows the focus area in Figure 7. Here, the main slick is 3–4 m wide, with the individual streamers separated on distance scales of ~1-m. Of these streamers, only the southernmost ($s_1$) corresponds to warm $\Delta T_B$, indicating thick oil, consistent with the bright patchy specular reflection, which also indicates likely thick emulsions. In contrast, the most northernmost streamer ($s_4$) corresponds to the cool continuous $\Delta T_B$ structure in the thermal infrared and thus is a thin sheen with $h < h_T$.

Notably, the thinner slicks are on the streamer's shoreward (downwind for sea breeze) side. The other two interior slicks are continuous in the visible but highly discontinuous in the thermal infrared. Given that thermal structures on the order of 1–2 pixels are well resolved, the discontinuities in the thermal could relate to slicks $s_2$ and $s_3$ being narrower than $s_1$ and $s_4$, with the cool slick ($s_4$), which is thinner, also being broader—this is expected for thinner oil. In addition, oil bunches on the slick's upwind side and spreads on the downwind.

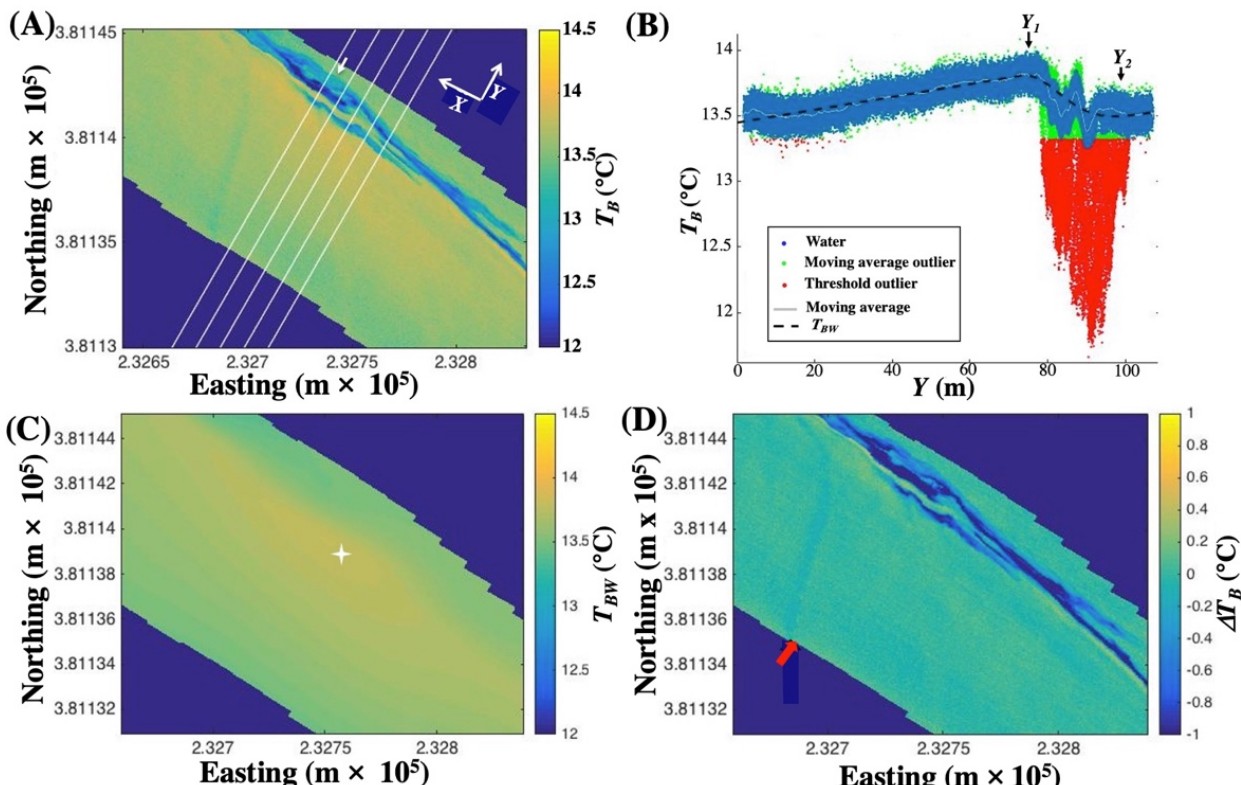

**Figure 7.** (**A**) Sea surface brightness temperature, $T_B$, mosaic. White arrow locates image for profile in Panel (**B**) and Figure 8. White lines delineate individual mosaic images. Along-slick, *X*, and cross-slick, *Y*, coordinates shown. (**B**) $T_B$ versus *Y* for image subset (white arrow Figure 8A), moving average, and model. (**C**) Background water $T_B$ and $T_{BW}$. White star locates warm water pool center. (**D**) Brightness temperature contrast, $\Delta T_B$. Red arrow shows boat wake, data keys on panels. Imagery acquired 23 May 2016.

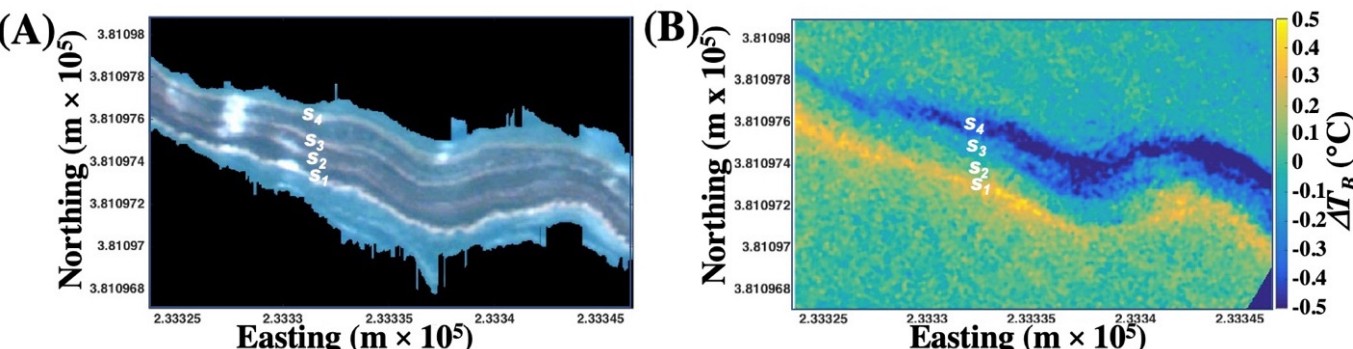

**Figure 8.** Focus area (**A**) Visible slick image and (**B**) Thermal brightness contrast, $\Delta T_B$. Streamer internal slicks labeled. Scene location shown by white arrow in Figure 7A. Data keys on panels imagery acquired 23 May 2016.

SeaSpires imaged a second streamer on 25 May 2016 (Figure 9). This slick exhibited a gradual kilometer-scale curvature, whereas streamer curvatures on 23 May were 250-m scale and included prominent structures on far smaller scales, tens of meters. Similar to the 25th, the water inshore of the slick is cooler than offshore, with the pooling of warm water along the slick trajectory also evident.

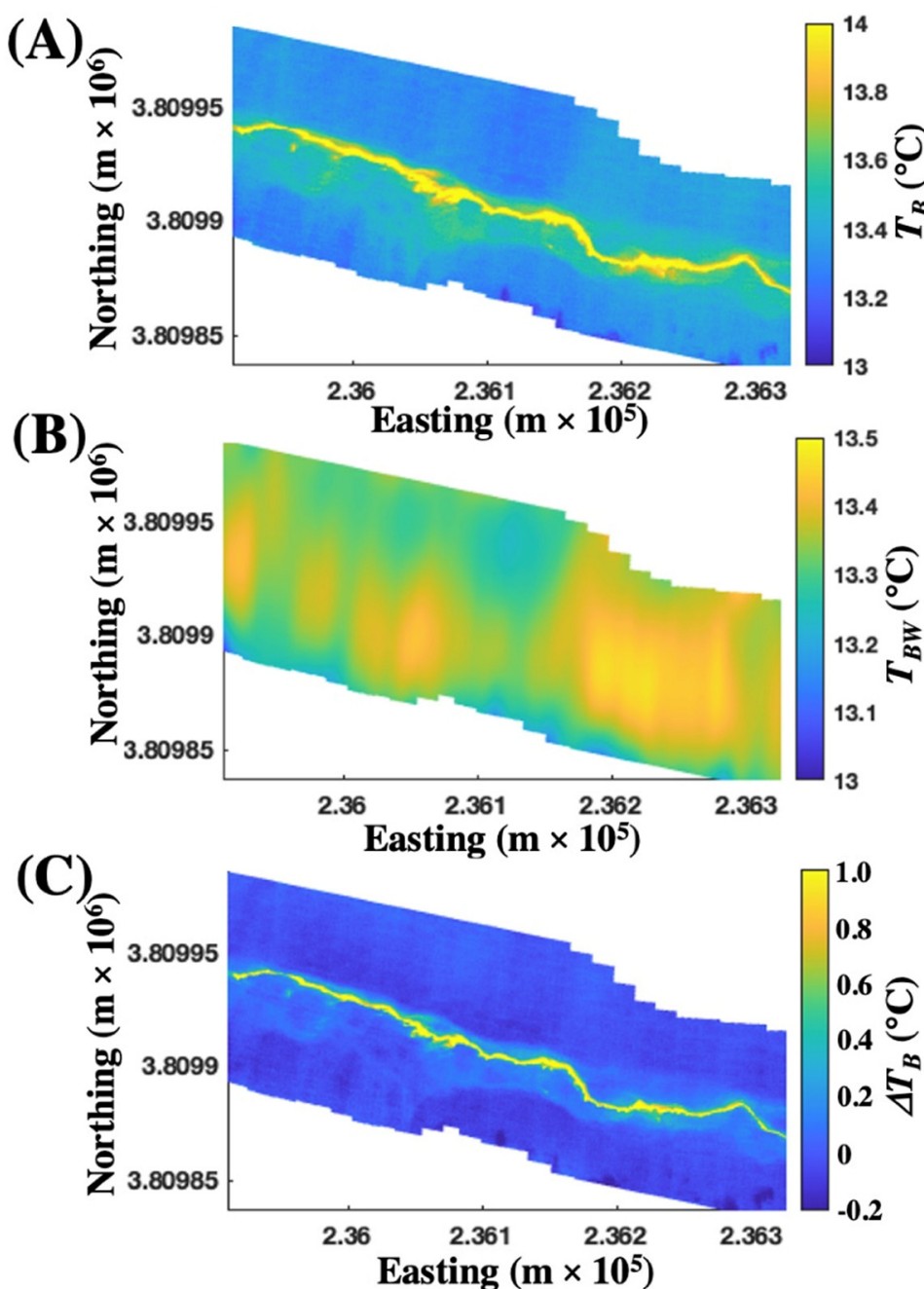

**Figure 9.** (**A**) Visible image photomosaic of an oil slick for 25 May 2016 oriented in the along-slick and cross-slick coordinates, $X$ and $Y$, respectively. (**B**) Mosaicked thermal infrared image of sea surface brightness temperature, $T_B$, projected to a 20-cm grid. (**C**) Water brightness temperature ($T_{BW}$) map. Data key on the figure.

Thermal structures in $T_{BW}$ were less pronounced on 25 May and also less associated with meanders—the structure at $X$ = 525–675 m was an exception (Figure 10). There was also no evident thermal discontinuity across the slick, suggesting this slick was not trapped by the main eddy west of COP (Figure 4).

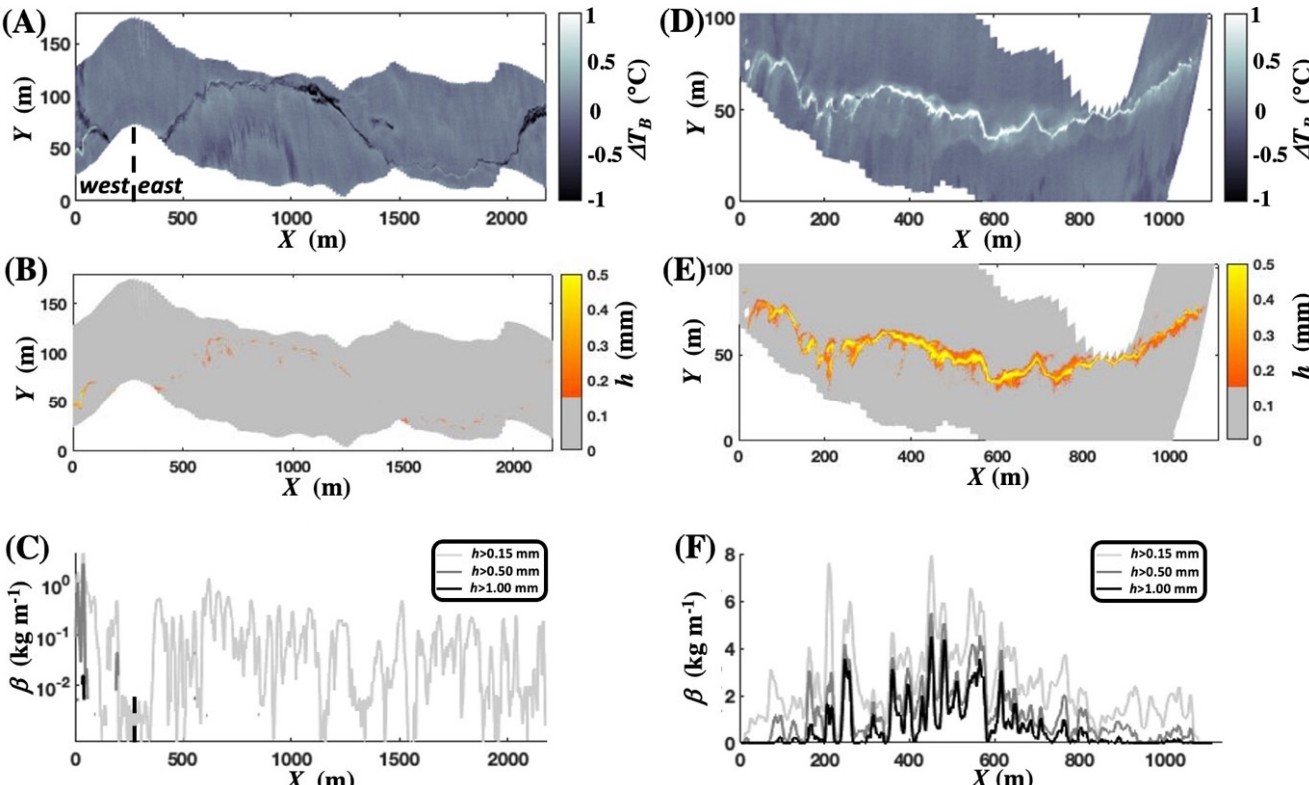

**Figure 10.** (**A**) Oil brightness temperature contrast, $\Delta T_B$, map, (**B**) slick thickness, $h$, map based on empirical calibration (east and west slicks labeled were considered distinct), and (**C**) along-slick linear mass ($\beta$), by integrating $\int h(X, Y) \, dY$, where $X$ and $Y$ are along-slick and cross-slick distances, respectively, for imagery acquired 23 May 2016. (**D**) $\Delta T_B$ map, (**E**) $h$ map and (**F**) $\beta$ for imagery acquired 25 May 2016. Data keys on panels.

### 3.5. Floating Slick Oil Mass

Application of the empirical model (Figure 5) to $\Delta T_B(X, Y)$ yields oil thickness maps, $h(X, Y)$ with the oil slick linear oil mass, $\beta$, (kg m$^{-1}$) calculated by $\int h(X, Y) \, dY$ across the slick (Figure 10C,F). Thermal structures unrelated to oil were below the 0.15-mm thick-oil cutoff, for example, the cool, half-arc feature ~75 m south (offshore) of the slick ($700 < X < 1000$ m). The slick on 23 May was considered as two different slicks, an east slick and a west slick. Whereas the cool and warm oil slicks follow each other for the east slick, the two deviate for the west slick.

The spatial heterogeneity was far more variable for the 23 May slick than the 25 May slick, repeatedly dropping to the noise level, with the vast majority of oil in the $h$ range 0.15–0.5 mm except for the west slick on 23 May. In contrast, the 25 May slick was far more persistent with far higher $\beta$, and a trend that generally increased and then decreased in the imagery. There were several gaps where the slick largely disappeared, e.g., $X = 375$ m and $X = 220$ m; although the latter corresponded to significant deformation of the slick orientation, suggesting a transport-related gap rather than related to a pause in emissions.

Some variations in $\beta$ appear repetitive, suggesting preferential timescales. A time-varying spectral analysis investigated scales in $\beta$ (spectrogram.m, Mathworks, MATLAB, Nantick, MA, USA). The background sea surface temperature showed more fine-scale structure on 25 May than on 23 May, with several focused structures on spatial length scales, $\chi$, ~50–100 m (Figure 11A,B). Short $\chi$ variations in $\beta$, ~10–50 m (i.e., short timescale variations), were similar in importance (power) on both days despite the differences in the background spectrograms power, $P$, at shorter $\chi$ (10–100 m).

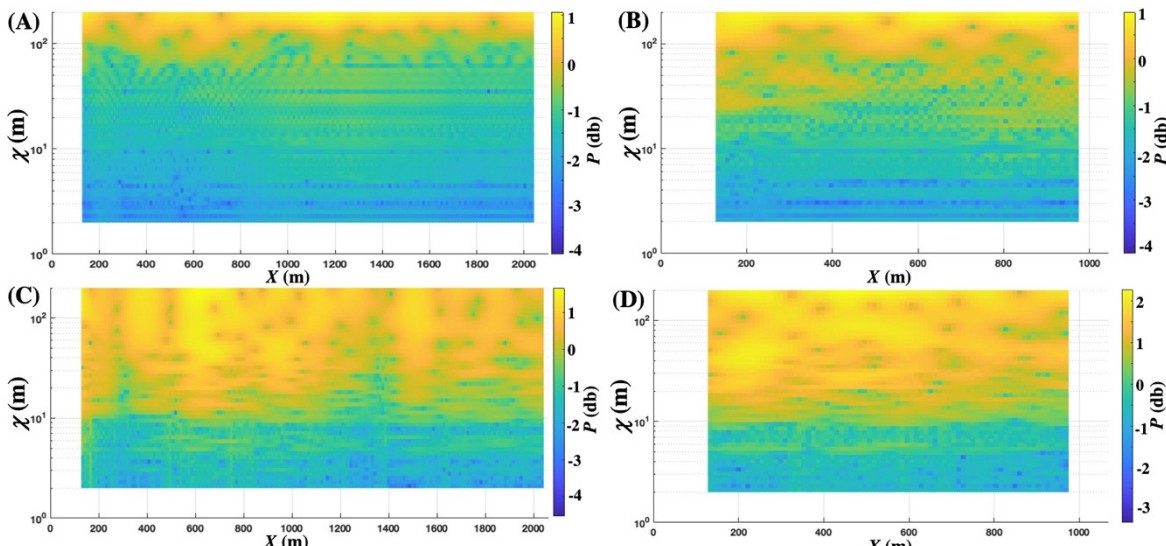

**Figure 11.** Power, $P$, spectrograms for sea surface background temperature of length scale, $\chi$, versus along-slick distance, $X$, for (**A**) 23 May 2016 and (**B**) 25 May 2016 and $\chi$ for oil streamer along-slick linear mass, $\beta$, versus $X$ for (**C**) 23 May 20161 and (**D**) 25 May 2016. Data keys on panels.

One significant difference in the $\beta$ spectrograms was that structures on 25 May were more persistent than on 23 May, when there were many short-lived events with significant $P$ from ~10 < $\chi$ < ~200 m. The poor correlation between the background spectrograms and their respective $\beta$ spectrograms suggest that slick spatial variability was more the result of emissions variability than transport variability except at the largest spatial scales. At the largest scales, there is an obvious visible correlation in the maps (Figures 6 and 9); for example, on 25 May, warm water pools at 200, 600, and 800 m, where the spectrograms show localized, elevated $P$ across a broad range of $\chi$. Similar patterns also are evident on 23 May.

The 23 May west slick was classified as distinct from the 23 May east slick. The $\Delta T_B$ probability distribution, $\psi$, supported this distinction, given the similar $\psi$ for the 25 May and 23 May east slicks (Figure 12). In contrast, $\psi$ on the 23 May for the west slick decreased far more steeply with $\Delta T_B$. The 25 May extended to far warmer $\Delta T_B$ (and thicker $h$) and comprised significantly more oil.

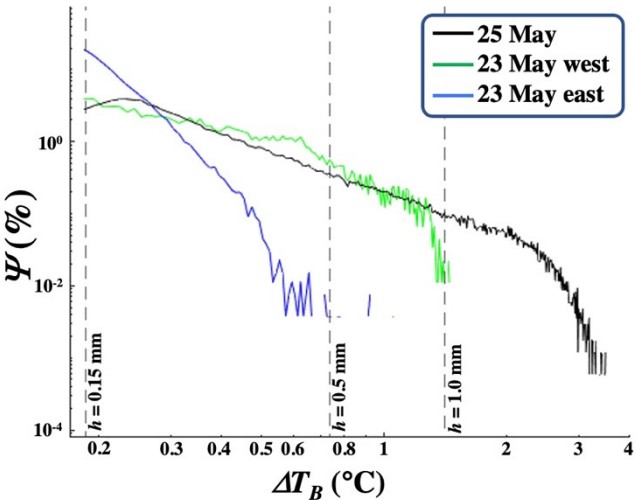

**Figure 12.** Brightness temperature contrast, $\Delta T_B$, probability distribution, $\psi$, for 23 and 25 May for several cutoff thicknesses, $h$, shown by dashed vertical lines. Data key on figure. See Figure 10A for the east and west slicks on 23 May.

The probability distribution of linear floating oil, $\Phi$, showed a broad-peaked distribution, with the thickest oil decreasing by a power law of $-3.27$ ($R^2 = 0.99$) (Figure 13). The thin oil contribution to the slick was negligible on 25 May, whereas it was dominant for the 23 May west slicks and very important for the 23 May east slick (thin oil is defined as $h > 0.15$ mm). $\beta$ for the 23 May west slick exhibited three modes centered at 1.0, 2.2, and 9.1 kg m$^{-1}$.

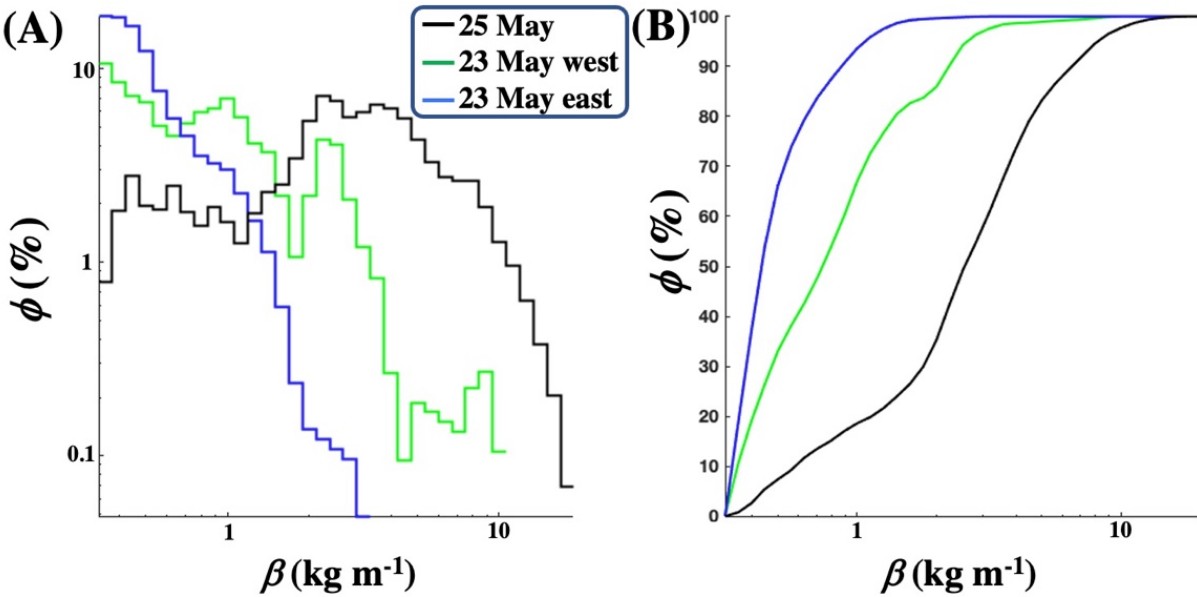

**Figure 13.** (**A**) Slick linear oil mass, $\beta$, (kg m$^{-1}$) normalized probability distribution, $\Phi$, (normalized to kg m$^{-1}$ bin$^{-1}$) for oil streamers from 23 May and 25 May. Data key on figure. See Figure 10A for the east and west slicks on 23 May. (**B**) Cumulative plot of $\beta$.

Most (61%) of the oil was contributed by moderate emissions, $3 < \beta < 25$ kg m$^{-1}$, with only 3.4% contributed by heavy emissions. Light emissions contributed the remaining 39%. These distributions could not be modeled as either Gaussian or power-law functions.

Total floating oil in the slick surveys on 23 May and 25 May was 311 and 2670 kg, for surveys of 2.2 km versus 1.1 km, respectively. Uncertainties were 25% and 7% on 23 May and 25 May, respectively (Supplementary Figure S6). Consideration of $\beta$ eliminates the measurement effect of survey length and was highly distinct between the 23 May and 25 May slicks, 0.14 versus 2.4 kg m$^{-1}$, respectively.

Emissions Estimate

Based on the drift velocity, the along-slick direction spatial coordinate corresponds to the time coordinate in terms of emission rate. To derive emissions from $\beta$ requires the interfacial speed over the period of drift from the surfacing location to its remote sensing observation location. The experimental plan was to use linearly extrapolated CODAR surface current data corrected to the interfacial drift velocity based on the microsphere drifter data. However, CODAR data begins approximately 3-km offshore and thus requires extrapolation to the more inshore location of the oil slicks. This extrapolation was infeasible due to the eddy from COP (Figure 4C), which means inshore currents west of COP are not an extension of offshore currents.

Since CODAR currents could not be used directly, loose guidance on converting X to emissions was based on the probability distribution of current speeds in CODAR data. The most common (median) current speed near COP for May 2016 was 0.2 m s$^{-1}$, implying a 1-km slick corresponds to 80 min. The mean speed was 23 cm s$^{-1}$ with a standard deviation of 14 cm s$^{-1}$.

Of course, velocities could be larger or smaller; however, we convert floating oil to emissions using this value as a discussion point. Specifically, the 23 May streamer

of 300 kg/2.2 km corresponds to ~30 g s$^{-1}$, and the 25 May streamer at 2700kg/1.1 km corresponds to 500 g s$^{-1}$. The average of these two streamers suggests single streamer emissions of ~8 kton yr$^{-1}$ (~200 bbl day$^{-1}$) within 2 to 14 kton yr$^{-1}$ (60 to 340 bbl day$^{-1}$) based on applying the standard deviation of the current speed to the median current speed. This estimate excludes thin (cool $\Delta T_B$) oil—a conservative bias.

Streamers correspond to focused seep areas in terms of oil emissions. There are approximately four continuously active major, oily seep areas (Figure 4, La Goleta Seep, Seep Tent Seep, Trilogy Seeps, Horseshoe Seeps) with additional intermittently active seep areas from shallower waters off of the Isla Vista coast. Typically, there are four to five streamers from the COP seep field (Leifer, Personal Observations, 2016). Thus, even if currents for the imagery were significantly higher than 0.2 m s$^{-1}$, these findings suggest that total field emissions likely are significantly greater than 200 bbl day$^{-1}$.

## 4. Discussion

### 4.1. Approach

There is a critical oil spill response need for oil thickness maps, yet quantitative (rather than semi-quantitative or simply qualitative) operational oil thickness remote sensing remains lacking; also lacking are field validation and quantification of methodology uncertainty, for which there are several underlying reasons. Firstly, marine oil spill conditions (illumination, winds, currents) for remote sensing are not feasible to approximate in the laboratory. Secondly, planned releases are extremely challenging to impossible to permit and suffer from magnitude and persistence limitations and potential weather interferences during the campaign—with none approved since 1990 in the US [56]. Finally, the priority is spill mitigation, not methodology testing during an oil spill. This study demonstrates how leveraging natural marine hydrocarbon seepage as in the COP seep field, where real-world thick oil slicks are a persistent feature, overcomes these challenges.

The study design included simultaneous remote sensing of a segment of an oil slick streamer that was collected. There were small disagreements between the empirical model (Figure 5) and the underlying data points. These likely relate to differing solar insolation and air-sea temperatures between the collects. Correction for these variations (see future work, Section 4.5.) should improve the empirical model; however, Monte Carlo uncertainty simulations suggest a weak sensitivity for the range of conditions spanned by these collects.

Capture was by a weir skimmer and harbor boom, modified for application in far rougher seas than typical for oil recovery. This skimmer's performance provided more time in the field. There were many challenges in coordinating three boats and an airplane to function effectively as a team. Many factors could compromise a collection—if either boom boat motors too fast, significant oil may be injected into the water, leading to collect abandonment. The boats gently eased the boom into the oil streamer—in and out of gear while also maintaining heading and orientation under the turning influence of winds and currents. This process was complicated, and abandoned collects outnumbered successful collects. Additional complexity arose from air traffic control as some of the COP seep field lies in the approach pathway of the Santa Barbara airport.

The study used airborne achromatic thermal contrast remote sensing to quantify oil thickness and thus does not leverage TIR spectral features. It is thus susceptible to false positives and negatives [31]. Combining multiple approaches by a semi-supervised pixel classification improves accuracy [57] and was implemented in this study for TIR and visible remote sensing of the collects.

This study focused on remote sensing of $\Delta T_B$ for thick oil, i.e., actionable, which was converted into *h* by an empirical model based on an in-scene calibration. The use of $\Delta T_B$ avoids the need for an emissivity correction, which would require correctly classifying thin oil whose $\Delta T_B \sim 0\,°C$ as distinct from oil-free sea surface for which $\Delta T_B = 0\,°C$ by definition. This issue is discussed in future research needs below.

This analysis presumes equilibrium—i.e., the heat flux is in steady-state. In reality, illumination, oil thickness, and environmental conditions continuously change. As such,

the relevant time scales must be considered to determine whether the equilibrium or steady-state assumption is appropriate. These vary from turbulence fluctuations on seconds to sub-second time scales and diurnal cycles on hourly time scales. Given that oil slicks are generally several millimeters to sub-millimeter, equilibrium likely is a good assumption for diurnal time scales; likely a poor assumption at turbulence time scales.

### 4.2. Sea Surface Thermal Structure

$\Delta T_B$ was calculated relative to the oil-free sea surface $T_B$, which exhibits thermal gradients driven by oceanographic processes. Thus, $\Delta T_B$ was calculated relative to a modeled sea surface $T_B$ based on sea surface $T_B$ outside the oil. Note, the presence of oil has positive feedback by elevating the sea surface and near-surface water temperature, although overall solar heating of upper ocean waters—incident energy is the same—is less due to greater radiative and convective losses to the atmosphere are greater for an oil surface. Typical oil-free surface $\Delta T_B$ gradients were ~0.01K m$^{-1}$.

The *SST* was discontinuous across the oil slick where the water inshore of the slick was cooler than water offshore of the slick. The presence of the slick at the *SST* discontinuity is not coincidental—absent sufficiently strong crosswinds, oil slick streamers aggregate at convergence zones, downwelling regions, or shear zones. When winds increase later in the day (typically with the onset of the sea breeze), trapped slicks may disconnect from the current or eddy feature, as for the 25 May slick (Figure 10).

A persistent inshore counter-circulation eddy to the west of COP is an important controlling factor in the location and fate of COP seep field oil slicks, illustrated by the Mako sea surface $T_B$ map (Figure 4). This eddy lies inshore of the CODAR data and drives surface currents parallel to the shoreline to the west of COP and then towards shore—a pattern documented in drifter studies [55]. The cooler water in the eddy's center suggests upwelling and divergence, which by continuity drives outwelling and convergence at the eddy's edges, i.e., the eddy is associated with three-dimensional fluid motions.

The importance of Lagrangian Coherent Structures (LCS) as an emergent fluid transport property that affects oil slick transport is well-recognized [58], directing oil slicks. Although LCS are insensitive to isolated and short-lived velocity field perturbations, they are affected by persistent wind fields—described in oil transport models by windage [59], which modifies (stretches and narrows) the LCS (at the surface). Offshore near-coast winds favor upwelling, whereas onshore winds favor downwelling. These wind-driven motions affect the upper water velocity field and are distinct from deeper currents [59].

The eddy directs oil slicks along the coast and then towards the coast several kilometers west of COP, with surface fluid motions in the current shear aggregating oil. As this oil drifts, it weathers until the slick's density exceeds seawater's density and the oil disperses into the upper water column (Leifer, personal observation, 2016). Thus, eddy transport time scales compete with weathering time scales to determine whether the tar sinks or beaches.

Typical northern Santa Barbara Channel diurnal current and wind patterns are strong prevailing winds from the west that pick up late morning and persist strongly through the late evening, sometimes past midnight. Although the bulk water-column current flows westwards, these winds drive surface currents eastward when sufficiently strong [14]. When prevailing winds dominate over gyre oil transport, the slicks move towards COP [54] with higher beach tar near COP than further westwards beaches [14]. This suggests oil dispersion during gyre transport is significant, mitigating this transport pathway to beaches west of COP.

The remote sensing TIR imagery reveals fine-scale spatial thermal structure in the eddy west of COP, likely related to shear instabilities at the eddy's edge (Figure 4). Instabilities in a sheared current flow are expected [60]. Specifically, both Mako data and the along-slick SeaSpires $T_B$ show along-slick direction *SST* variations. For example, SeaSpires $T_B$ data show warmer water pools at multiple locations, one example being adjacent to the large oil aggregation at $x$ = 1100 m on 23 May (Figure 6C). Underlying oceanographic motions also drive meanders in the oil slick trajectories (Figure 10).

Positive feedback may play a role—oil slick absorption of solar energy will tend to increase near-surface water-column stability compared to the nearby oil-free sea surface, leading to weakened vertical mixing with cooler, deeper water. Moreover, mixing is less in the convergence flow at the eddy's edge, where water outside the eddy is warmer in proximity to the eddy's edge (Figure 4B). This phenomenon was observed in the warm water band adjacent to the oil slick (Figure 7B). This band of warm water extends beyond the main thick oil slick, even in areas with no visible slicks, e.g., $X = 675$ m, and thus cannot be solely a thin slick emissivity effect.

### 4.3. Oil Slick Thermal Structure

For thick oil, even strong (~0.3 °C in Figure 7) spatial *SST* structure from oceanography did not challenge the analysis focused on thick oil (large $\Delta T_B$), although modeling $\Delta T_{BW}$ was critical to calculating an accurate $\Delta T_B$. Additionally, thick oil tended to have sharp edges or transitions and was highly localized spatially on a meter and smaller scales. In contrast, thermal structures corresponding to thin oil (e.g., ~0.25 °C, Figure 6, $X = 850$ m, $Y = 50$ m) were gradual and thus shared similarity with oil-free seawater thermal structures.

The thermal imagery (15-cm resolution) characterized the dominant internal slick thermal structures. Specifically, the major slick thermal structures and sub-structures are resolved as spanning multiple pixels in Figure 8, even for structures with $\Delta T_B < 0.1$ °C. Visible imagery shows an even finer spatial structure, some of which was replicated only poorly in the thermal imagery, primarily for thinner oil slicks. Specifically, thinner slicks in the visible likely were discontinuous in the thermal (Figure 8). This discontinuity likely arises from sub-pixel heterogeneity, with the narrower internal streamers containing a greater fraction of thin oil than the wider internal streamers (from outside the streamer and from within). The non-linear response of $h$ to $\Delta T_B$ (Figure 5) magnifies this effect. Other possible explanations for the lack of equivalently fine-scale TIR structure is crosstalk between adjacent pixels in the sensor or lateral heat transfer, smoothing *SST* gradients.

### 4.4. Slick Time History and Unsteadiness Implications

Spatial variations in the along-slick direction reflect the time history of oil emissions with the conversion between distance and time based on currents. Extrapolation of CODAR velocities (which extend to only ~3 km offshore [61]) to inshore waters where the slicks were located was infeasible due to the persistent clockwise re-circulation eddy west of COP. Thus, typical high and low current CODAR speeds for May 2016 were used to constrain drift times and hence emissions time history.

The derived floating oil mass, $\beta$, for the two streamers demonstrated two distinct emissions characteristics. Short-term variability (~20 m) could relate to transport processes or emissions variability. Specifically, whereas the emissions feeding the 25 May streamer were far steadier, the 23 May streamer was characterized by sporadic with large transient emissions separated by periods of low and relatively stable emissions. Notably, transient emissions are a significant component of the overall emissions and are more than a magnitude greater than the quasi-steady state emissions. Shorter length-scale structures likely arise from sea surface flow structures, highlighted in the brightness temperature maps in Figure 4C.

For several of the 23 May high mass aggregations, $\beta$, the rise is faster than the decrease, for example, at $x = 600$, 1100 m, and 1450 m (Figure 10). This pattern of emissions was documented for several large gas transient emissions [62–64]. Leifer, Luyendyk, Boles and Leiferet al. [63] proposed that such eruptions relate to the failure of a temporary seal of the primary migration pathway(s), with the seal cleared by the eruptive event. In this case, the high viscosity of oil in the migration pathways likely causes the blockage. As the oil is pushed out of the pathways, gas emissions increase. Shifting between oil and gas was observed for an abandoned oil well offshore California, albeit non-eruptively, identified as slug flow [27]. Slug flow is a flow that alternates between mostly gas and then mostly liquid phases.

The estimated streamer oil emission was ~200 bbl day$^{-1}$ based on a typical (median) current of 20 cm s$^{-1}$. Given that the seep field typically emits four or five streamers, total emissions are likely significantly larger, notably, significantly higher than the reported seep field emissions of 100 bbl day$^{-1}$ for the mid-1990s by Hornafius, Quigley and Luyendyk [22]. Hornafius, Quigley and Luyendyk [22] estimated seep bubble emissions from a boat-based survey for depths >15 m (missing the shallowest seepage) and applied the oil to gas ratio from the seep tent to the field. Thus, higher oil emissions in 2016 than in the 1990s are consistent with higher gas emissions in 2016. Specifically, Bradley, Leifer and Roberts [24] found that 1994–1996 emissions (based on atmospheric concentration at a nearby onshore air quality station) were well below the 1990–2008 values, with significant increases after 2008. There are also seasonal trends to consider, which favor higher overall emissions—the Hornafius, Quigley and Luyendyk [22] surveys occurred in summer and fall when seepage activity is at a minimum, whereas winter and spring feature much higher activity associated with seasonal storms [24].

The uncertainty of this emissions estimate is significant—highlighted by order of magnitude differences in $\beta$ between the two streamers, uncertainty in the actual currents, and whether the characterized streamers are representative of other field streamers. Additional uncertainty arises from the unknown seasonality of oil emissions and transiency—the data herein showed multiple eruptions and highlighted the importance of unsteady emissions. Moreover, transiency is expected in oil and gas flows as slug flow.

### 4.5. Future Directions and Lessons Learned

This paper provided a basis for estimating floating thick oil from remote sensing data. Thin oil, which appears cold in $\Delta T_B$, was not included but should be addressed for several applications, such as emissions assessment. Thin oil requires an emissivity correction, which differs between oil and oil-free seawater. Given that $\Delta T_B = 0$ for seawater (by definition) and oil with thickness, $h$, near the transition thickness, $h_T$, pixel classification as oil or seawater is critical; thus, $\Delta T_B$ cannot be the basis of this classification. Assessment could be from visible imagery, a planned effort using the collected data.

The along-slick survey data characterized short-term variability; however, the time conversion was based on estimated rather than measured surface currents. Future SAS studies should include microsphere surface current mapping, e.g., Leifer, Luyendyk and Broderick [19], wherein the sea surface is seeded with hollow glass microsphere that can be GPS tracked by boat or airplane.

In this study, $\Delta T_B$ was derived from $\Delta T_B$ based on the in-scene calibration, with $\Delta T_B$ derived from the $T_B$ contrast of the oil with $T_B$ of oil-free seawater (which was modeled). This required thermal imagery of the sea surface beyond the oil slick. Although this generally was achieved, in several places, data outside the slick was minimal (Figure 6B). In one case, the slick lay completely outside the TIR camera field of view (though it remained in the visible camera field of view). One solution is to acquire imagery at two altitudes; another is to fly three or more flight lines in a raster pattern or add a second TIR camera with a wider field of view to characterize SST gradients better.

In many conditions, such as during an oil spill response, the in-scene calibration may be infeasible. Developing a theoretical framework would allow the extension of these field observations (Figure 1) to different scenes (insolation, environmental conditions) and oils. Specifically, a numerical heat model that includes radiative and heat transfer processes, including turbulence heat transfer. A numerical model can also provide uncertainty estimates for various conditions. Numerical models require validation, specifically comparison with detailed vertical thermal profile observations of floating oil layers of different thicknesses in the field and laboratory. Manuscripts on these efforts are under review. Furthermore, a numerical model would improve the empirical model by accounting for the effect on $\Delta T_B$ of different solar insolation and environmental conditions, i.e., the air-sea temperature difference between collects.

The SAS approach leveraged direct oil capture to field-verify oil spill remote sensing for oil from the COP seep field—a very low API crude. Based on theoretical considerations (Figure 1), correction factors can be calculated for application to other crude oils and emulsions based on a numerical radiative/heat transfer model.

## 5. Conclusions

The COP seep field provides an ideal natural laboratory to study oil slick processes. Natural seepage needs no permitting and, as a continuous release, does not exhibit startup dynamics and allows repetition under diverse weather conditions.

The SAS approach leveraged direct oil capture to field-verify oil spill remote sensing for oil from the COP seep field—a very low API crude. Based on theoretical considerations (Figure 1), correction factors can be calculated for application to other crude oils and emulsions based on a numerical radiative/heat transfer model.

**Supplementary Materials:** The following supporting information can be downloaded at: https://www.mdpi.com/article/10.3390/rs14122813/s1. Supporting information including figures and descriptions, visualization of currents, details on methodology of determining thermal contrast. Reference [65] is cited in the supplementary materials.

**Author Contributions:** Conceptualization, I.L. and D.M.T.; methodology, I.L. and D.M.T.; software, I.L., C.M. (Christopher Melton), J.D.K., P.D.J. and K.N.B.; formal analysis, I.L., D.M.T., C.M. (Christopher Melton), J.D.K., P.D.J. and K.N.B.; validation, I.L. and D.M.T.; visualization, I.L., C.M. (Christopher Melton), P.D.J. and K.N.B.; writing, review, and editing, All; original draft preparation, I.L. and D.M.T.; acquiring funding, I.L. and D.M.T.; administration, I.L. and D.M.T. All authors have read and agreed to the published version of the manuscript.

**Funding:** This research was funded by Plains All American Pipeline and The Aerospace Corporation's Independent Research and Development program.

**Data Availability Statement:** The data presented in this study are available on request form the corresponding author. The data are not publicly available de to legal concerns.

**Acknowledgments:** We gratefully acknowledge the support of Plains All American Pipeline for the SAS study. We also thank Marc Mortisch and Joel Cordes, Santa Barbara Air Pollution Control District, for providing the West Campus Station data. Mako imagery was acquired under the auspices of The Aerospace Corporation's Independent Research and Development program. The knowledge, coordination, and seamanship of Gordon Cota are gratefully acknowledged for his key contribution to the success of the SAS. Bill Behrenbruch's (Visual Systems) efforts for piloting the airplane are acknowledged. Finally, the contribution and excellent seamanship of vessel captains Jeff Wright (F/V Double Bogey), Fred Hepp (F/V Gloria Maria), and Tony Vultaggio (F/V Rock Steady), John Colgate (F/V Maalea). A special acknowledgment is made in memorial to Ron Fairbanks (F/V Devin), who could not participate through the study's end.

**Conflicts of Interest:** The funding sponsors has no role in the design of the study, in the collection, analyses, or interpretation of data; in the writing of the manuscript and in the decision to publish the results. The authors declare no conflict of interest.

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
