# Peer review of "Measuring Floating Thick Seep Oil from the Coal Oil Point Marine Hydrocarbon Seep Field by Quantitative Thermal Oil Slick Remote Sensing"

_remotesensing, doi:10.3390/rs14122813_

Round 1

Reviewer 1 Report

This paper describes the results of an experiment designed to calibrate thermal infrared estimates of oil thickness. The paper then uses that information to estimate the amount of oil being released from the Coil Oil Point natural seeps. The paper seems to stray away from being focused on these two items, which makes it more difficult to follow. There is analysis of spectra collected in a different year with a different system. There is a lot of discussion on oil weathering and circulation patterns that doesn’t seem connected with the main points. There are several sections where the grammar needs to be improved to assist the ability of a reader to follow the paper. While generally, a very detailed theoretical analysis is provided, I was not able to find a discussion on the importance of reaching thermal equilibrium and how that may depend on the solar insulation. This concept is important in understanding if the calibration would likely change through the day or between days. There are very few calibration points shown in Figure 5; however, the data from the two different days lay on opposite sides of the calibration curve. This may be a result of differences in solar insolation.

I think there is value in the data and techniques presented. The paper would benefit from a reduction in materials not needed for the primary purposes and another read for grammar. Some discussion of how the results may depend on the oil reaching thermal equilibrium should be included.

Section 1.4 Why is this section needed? Consider removing.

Line 152-155 The energy flows are time dependent. They depend on the solar insolation history to determine the heating of the oil. There should be a mention of the time dependent component beyond the reference on line 189.

Line 247 and section 2.3.7 I am not sure why this is included in this paper and recommend deleting it.

Line 333 You introduce the y variable without defining it. Please define.

Line 380 If you keep the hyperspectral discussion correct the colors indicated on this line.

Line 623 I don’t understand how the longer survey on the 23rd led to a much smaller slick being observed. Are the difference really related to difference in survey length?

Line 709-742 I don’t see the relevance and suggest deleting.

Section 4.4 I am not sure why we need to understand the seep dynamics at this detail.

Line 811 “… higher current gas emissions consistent with significantly higher current gas emissions.” This sentence needs to be clarified.

Lines 837-843. Please explain why this is important. I can come up with a variety of reasons but it would be good to know what you are thinking.

Line 847 I was left wondering how well TB of oil free water was established if oil extended throughout the image. This should be addressed in the methods.

Author Response

Please find a point by point response in the attached file

Reviewer 2 Report

Manuscript "Measuring floating thick seep oil from the Coal Oil Point marine hydrocarbon seep field by quantitative thermal oil slick remote sensing" is devoted to the very important scientific and applied problem of determining oil film thickness from remotely sensed data. It presents an interesting method based on the use of infra-red data. The material presented in the paper is very detailed, sufficiently clear even for non-specialists in the field.
The paper is interestingly structured, even the introduction is divided into sub-paragraphs. Unusual, but interesting.
The manuscript can be accepted in present form.

Author Response

We are in agreement with this review.

Reviewer 3 Report

  • The authors present a very interesting approach to leverage oil emissions from the Coal Oil Point (COP) natural marine hydrocarbon seepage offshore of southern California, where prolific oil seepage produces thick oil slicks stretching many kilometers. They demonstrated and validated a remote sensing approach as part of the Seep Assessment Study (SAS). The manuscript is clear, relevant for the field and presented in a well-structured manner and scientifically sound. The manuscript’s results are reproducible based on the details given in the methods section. One minor remark: Authors should mention more about their future work. 

Author Response

We have added a paragraph in section 4.6, detailing “future work” including a numerical heat model, and validation studies in the laboratory and field, in three manuscripts, with the lab study currently under review, the others ready for submission pending acceptance of the lab study paper. As such, we have not cited them, though welcome editor feedback on this matter.